# Therapeutic Potential of Flavonoids and Tannins in Management of Oral Infectious Diseases—A Review

**DOI:** 10.3390/molecules28010158

**Published:** 2022-12-24

**Authors:** Ján Kováč, Lívia Slobodníková, Eva Trajčíková, Katarína Rendeková, Pavel Mučaji, Alice Sychrová, Silvia Bittner Fialová

**Affiliations:** 1Department of Stomatology and Maxillofacial Surgery, Faculty of Medicine, Comenius University in Bratislava, Heydukova 10, 812 50 Bratislava, Slovakia; 2Department of Stomatology and Maxillofacial Surgery, St. Elizabeth’s Hospital, Heydukova 10, 812 50 Bratislava, Slovakia; 3Institute of Microbiology, Faculty of Medicine and the University Hospital in Bratislava, Comenius University in Bratislava, Sasinkova 4, 811 08 Bratislava, Slovakia; 4Department of Pharmacognosy and Botany, Faculty of Pharmacy, Comenius University in Bratislava, Odbojárov 10, 832 32 Bratislava, Slovakia; 5Department of Natural Drugs, Faculty of Pharmacy, Masaryk University, Palackého 1946/1, 612 00 Brno, Czech Republic

**Keywords:** antimicrobial activity, antibiofilm activity, flavonoids, medicinal plants, natural products, oral infections, oral pathogens, tannins

## Abstract

Medicinal plants are rich sources of valuable molecules with various profitable biological effects, including antimicrobial activity. The advantages of herbal products are their effectiveness, relative safety based on research or extended traditional use, and accessibility without prescription. Extensive and irrational usage of antibiotics since their discovery in 1928 has led to the increasing expiration of their effectiveness due to antibacterial resistance. Now, medical research is facing a big and challenging mission to find effective and safe antimicrobial therapies to replace inactive drugs. Over the years, one of the research fields that remained the most available is the area of natural products: medicinal plants and their metabolites, which could serve as active substances to fight against microbes or be considered as models in drug design. This review presents selected flavonoids (such as apigenin, quercetin, kaempferol, kurarinone, and morin) and tannins (including oligomeric proanthocyanidins, gallotannins, ellagitannins, catechins, and epigallocatechin gallate), but also medicinal plants rich in these compounds as potential therapeutic agents in oral infectious diseases based on traditional usages such as *Agrimonia eupatoria* L., *Hamamelis virginiana* L., *Matricaria chamomilla* L., *Vaccinium myrtillus* L., *Quercus robur* L., *Rosa gallica* L., *Rubus idaeus* L., or *Potentilla erecta* (L.). Some of the presented compounds and extracts are already successfully used to maintain oral health, as the main or additive ingredient of toothpastes or mouthwashes. Others are promising for further research or future applications.

## 1. Introduction

According to data from the World Health Organization, oral diseases affect nearly 3.5 billion people throughout their lifetime, causing pain, discomfort, disfigurement, and even death and posing a significant health burden for many countries [1]. Most oral microbial diseases are polyfactorial in nature and have endogenous origins, with imbalanced oral microbiota being the main contributing factor [2] (Table 1).

### 1.1. Oral Cavity Pathogens

Dental caries, one of the most common infectious diseases worldwide, is a result of the cariogenic activity of a complex, diverse and dynamic microbial community of biofilm on the teeth surface—a dental plaque. In contrast with dental plaque compatible with oral health, the changed microenvironments in cariogenic dental plaque favor the expression of saccharolytic phenotypes and select for cariogenic pathogens (Table 2). Acidogenicity (ability to produce acids during sugar metabolism) and acidurity (ability to metabolize and multiply in acidic conditions) are the most important virulence factors of cariogenic microorganisms. Acids produced from dietary sugars attack the enamel and mineral components of dentin [10]. Once the primary dental lesion is established, the community of dental plaque bacteria with their degradative enzymes, toxins, and proinflammatory activities, contribute to dental caries spreading, the destruction of pulp, and eventual production of periapical granuloma or abscess, potentially fulminating in odontogenic sepsis and endangering the life of the patient [7]. Some non-fastidious microorganisms from oral microbiota can survive harsh conditions in the repaired dental canal and may start a chronic dental infection leading to root canal treatment failure (Table 2) [8].

Periodontal disease (gingivitis, periodontitis, pericoronitis, and periimplantitis) results from a cooperative activity of imbalanced subgingival dental plaque bacteria, expressing a proteolytic phenotype. Anaerobic bacteria are essential in pathogenesis (Table 2) [11]. The resulting damage to the periodontal tissue, dental ligaments, and the alveolar bone is based on the activity of bacterial aggressins, tissue invasion, and the pathogenetic role of the inflammatory response, including the release of host metalloproteinases [11,12].

Oral candidiasis is usually diagnosed in patients with decreased immunity and other underlying conditions and with oral microbiota disbalance (Table 1) [9]. The most frequent agent is *Candida albicans*, but many other species are frequently isolated (Table 2).

Oral microbiota-associated diseases have a local health impact and are sources of focal and disseminated infections (Table 1). Acute infectious complications are caused by microorganisms entering the bloodstream, leading to endocarditis, osteomyelitis, arthritis, or abscesses of parenchymatous organs. The spreading of dental infection *per continuitatem* may lead to severe head and neck infections, including life-threatening central nervous system infections [3]. Chronic periodontal disease maintains a persistent low-grade systemic inflammation, which can contribute to cardiovascular diseases, affect the process of diabetes, and is even connected to Alzheimer’s disease [4]. The current research data identified the imbalanced oral microbiota as a factor suspected of contributing to inflammatory bowel disease [5]. Moreover, considering halitosis and aesthetic defects, oral infections may lead to social drawbacks for the patient [6]. 

### 1.2. Natural Products in Oral Health

The treatment for oral health conditions is expensive, and many low- and middle-income countries cannot provide services to prevent and treat oral health conditions [1]. As cheap, accessible, and effective alternatives, medicinal plants and overall herbal products are very popular in the therapy of infections related to teeth and gums. Different herbal formulas for oral infections have been known from traditional medicines and are successfully used in current treatment [13]. A big advantage of natural remedies that intend to restore and support oral health is that they have various potential targets and complex activities. They can decrease the oral microbial burden, establish the equilibrium of the oral microbiota, and support bacterial communities compatible with oral health. Natural products, except to direct microbicidal activity on oropathogenic microorganisms, could decrease microbial virulence, including adhesivity, biofilm production, saccharolytic and proteolytic activity, or suppress microbial metabolism [14,15]. The biofilm-inhibiting, biofilm-eradicating, or quorum-quenching activity is also a wishful property of natural products included in the armamentarium of remedies to treat oral infectious diseases [16,17]. Anti-inflammatory, antioxidant, and immunomodulatory activities supporting healing and suppressing oxidation stress in periodontal tissue are similarly desirable properties. Moreover, an essential role in the management of periodontal disease may also be played by the inhibition of host metalloproteinases [12]. Pain relief, a decrease in gum bleeding, and the suppression of halitosis are all supportive interventions during periodontal disease therapy; protection and regeneration of enamel are required to prevent dental caries. Along with the therapy of underlying diseases, improved oral hygiene, and a change in diet and lifestyle, natural products with their various beneficial activities can be successfully included in the complex approach to therapy and the prevention of dental caries and periodontal disease [18,19]. 

European traditional medicinal plants, such as agrimony, marigold, witch hazel, rose, chamomile, oak, and many more, are intended by the European Medicines Agency (EMA) for oral infections and inflammations. The plants mentioned contain mainly polyphenols (flavonoids or tannins) as active compounds with antimicrobial, antioxidant, anti-inflammatory, and wound-healing effects [20]. However, traditional medicines in other parts of the world offer many different (specific or endemic) plants and natural products that are effectively used for oral infections and diseases. 

This paper aims to present two groups of polyphenols (flavonoids and tannins) that are active against microorganisms (bacteria and fungi), the causative agents of oral infections.

## 2. Results and Discussion

### 2.1. Flavonoids in Oral Health

Flavonoids, the most common and widely distributed phytochemicals in the plant kingdom, possess many beneficial biological activities, including antioxidant, anti-inflammatory, and antimicrobial [21,22,23]. Flavonoids can be divided into a variety of classes such as flavones (e.g., apigenin, luteolin), flavonols (e.g., quercetin, kaempferol, galangin), flavanones (e.g., hesperetin, naringenin), flavanonols (e.g., taxifolin), isoflavones (e.g., genistein, daidzein) and flavan-3-ols (e.g., catechin, epicatechin), which are precursors of tannins—catechins [24,25,26]. 

Flavonoids are synthesized as a plant response to microbial infection; thus, they are potent antimicrobial agents against a wide range of pathogenic microorganisms [27]. The primary step in the antibacterial activity of flavonoids is the interaction with the cell membrane (phospholipid bilayer). Whether the interaction occurs outside or inside the bilayer depends on the lipophilicity/hydrophilicity of the respective flavonoid [28]. Lipophilic substituents such as prenyl groups, alkylamino chains, alkyl chains, and nitrogen or oxygen-containing heterocyclic moieties in the flavonoid structure are a presumption for more potent antibacterial activity [28,29]. In summary, flavonoids possess antibacterial activity by different mechanisms: inhibition of nucleic acid synthesis, inhibition of energy metabolism, inhibition of the attachment and biofilm formation, inhibition of porins in the cell membrane, alteration of the membrane permeability, and attenuation of the pathogenicity [26,30]. Some flavonoids can reverse antibiotic resistance and enhance the effect of the current antibiotics [21]. Flavonoids may affect different bacterial enzymes such as proton translocating F-ATPases, which play significant roles in protecting bacteria against environmental stress caused by the acidification of biofilms [31], or sortase A (SrtA), a gram-positive bacterial membrane enzyme that contributes to virulence. SrtA could be found on the surface of *S. mutans* and is associated with the adhesion to host tissues, the evasion of host defenses, and biofilm formation. It is a well-known target for developing new anti-infective drugs [32,33]. One of the flavonoid molecular targets is the synthesis of extracellular glucans by glucosyltransferase (GTF), a proven virulence factor involved in caries pathogenesis. Glucans increase the pathogenic potential of dental plaque by promoting the adherence and accumulation of cariogenic streptococci on the tooth surface [34,35].

Flavonoids may be helpful in dentistry as prophylaxis against bacterial infections and plaque formation and as an adjuvant therapy to promote the postoperative healing of traumatized tissues in the oral cavity [26]. The most frequently tested flavonoids with antibacterial/antibiofilm effects in oral infections are summarized in Table 3.

Not only sole flavonoids, but also flavonoid rich medicinal plants and natural products showed potent antibacterial or antibiofilm action. 

Quercetin and kaempferol were identified as the most abundant compounds in Nidus vespae (honeycomb) extract, examined by Guan et al. (2012). Chloroform/methanol extracts of honeycomb inhibited the growth of various bacteria, including cariogenic (*S. mutans*, *S. sobrinus*, *S. sanguis*, *Actinomyces viscosus, A. naeslundii,* and *L. rhamnosus*), with MICs ranging from 1 to 4 mg/mL and MBCs from 4 to 16 mg/mL. At sub-MIC concentrations, they inhibited the acidogenicity and acidurity of *S. mutans* cells. In addition, the bacterial F-ATPase activity was reduced by 47.37% with 1 mg/mL of quercetin and by 49.66% with 0.5 mg/mL of kaempferol [48]. 

Apigenin, quercetin, luteolin, and their derivatives are the main flavonoids present in the aerial parts of *Matricaria chamomilla* L. [20,49]. The water extract of *M. chamomilla* exhibited anti-caries activity comparable to CHX, but the antibiofilm effect was relatively low [50]. In a randomized, double-blind clinical trial with 45 patients, *M. chamomilla* extract in Orabase protective paste reduced the pain of minor aphthous stomatitis [51]. 

Synergistic effect in the reduction of the dry weight of *S. mutans* biofilm and total amounts of extracellular insoluble glucans and intracellular polysaccharides occurred with the combination of *tt*-farnesol, fluoride, and myricetin. This combination also decreased the expression of glucosyltransferase B in biofilm [44]. 

Some flavonoids comprise lipophilic chain or chains of varying lengths in their molecule. These prenylated flavonoids revealed a very good antibacterial effect against gram-positive and gram-negative bacteria [52]. Furthermore, the prenyl group can react with the adjacent OH groups to form a heterocyclic ring. A molecular docking study identified prenylated flavonoids kurarinone and isobavachalcone among 178 tested compounds as the most potent inhibitors of *S. mutans* SrtA [33]. The prenylated flavonoids, including kurarinone, are contained in the roots of *Sophora flavescens* Aiton, a plant used in traditional Chinese medicine. The traditionally prepared root extract (water extraction followed by ethanol precipitation) and kurarinone exhibited strong antibacterial activity against *S. mutans,* with MIC = 16 μg/mL and 2 μg/mL, respectively. The activity of kurarinone was comparable to that of conventional antibiotics [53].

Badria and Zidan tested 39 compounds, including 17 flavonoids, against 4 strains of *S. mutans*. Unspecified flavone showed potent inhibitory activity with MIC 6.25 μg/mL with an inhibition of adherence ˃50%. Other flavonoids including apigenin, chrysin, 5,7-dihydroxyisoflavone, 3-hydroxyflavone, morin, myricetin, naringenin, quercetin, and rutin showed a moderate growth inhibitory activity at MIC 12.5 μg/mL. The MIC of the rest of the tested flavonoids was higher than 125 μg/mL [54]. 

Grape seed extract rich in flavonoids was tested on oral pathogens *F. nucleatum* and *P. gingivalis*. It exhibited a bacteriostatic effect at 2000 µg/mL and 4000 µg/mL, respectively, and significantly reduced the formation of biofilm [55].

Chilean researchers tested the antimicrobial activity of Chilean propolis (CEP), which is rich in flavonoids, on *S. mutans*. MICs of three samples of CEP were in the range of 0.22–0.91 μg/mL, and MBCs ranged from 0.91 to 1.30 μg/mL. The most abundant flavonoids in CEP, apigenin, pinocembrin, and quercetin, revealed antistreptococcal activity with MICs 1.3 µg/mL, 1.4 µg/mL, and 4.1 µg/mL, respectively. The MIC values of apigenin and pinocembrin obtained in this study were comparable to the MIC of CHX. In addition, these flavonoids can modify the structure of the *S. mutans* biofilm. The inhibition of biofilm formation and the reduction in thickness was observed [56,57]. Chilean propolis was tested also against other cariogenic bacteria, such as *S. sobrinus*. All propolis samples inhibited the growth of streptococci (MICs = 0.90 – 8.22 µg/mL). The HPLC/MS analysis of twenty propolis samples revealed the presence of flavonoids: quercetin, myricetin, kaempferol, rutin, pinocembrin, and phenolic acids: coumaric acid, caffeic acid, and caffeic acid phenethyl ester [58].

Flavonoids are very potent active substances in medicinal plants. Thanks to their antimicrobial and anti-inflammatory effects, the action of some flavonoids is comparable to that of conventional antibiotics. In this context, we can describe flavonoids with lipophilic substituents, such as prenylated flavonoids, as particularly effective. Many flavonoids can inhibit biofilm formation and suppress virulence factors. The structures of the most common flavonoids discussed in this work are figured in Figure 1. In addition, as the research shows, the combination of flavonoids with antibiotics can reduce antimicrobial resistance.

### 2.2. Tannins in Oral Health

Tannins are polyphenolic compounds commonly known for their astringent activity, which is responsible for a wide range of biological effects. The intense astringent feeling in the oral cavity during the consumption of foods rich in tannins is caused by the interaction of proteins in the oral mucosa and saliva with tannins, where the main actors are histidine parts of proteins. [59]. The biological activities of tannins are comparable with flavonoids. Tannins have a wide range of biologically interesting effects, including antibacterial and anti-inflammatory properties. Antibacterial activity is given by their astringent effect, which depends on the structure of the tannin molecule. Engels et al. (2011) state that a higher degree of galloylation and higher hydrophobicity results in stronger protein binding and a higher affinity for iron [60]. The antibacterial mechanism of the action of tannins includes (i) iron chelation, the (ii) inhibition of cell wall synthesis and disruption of the cell membrane, and the (iii) inhibition of fatty acid biosynthetic pathways. Tannins may also influence the gene expression of virulence factors (biofilms, enzymes, adhesins, motility, and toxins) and act as quorum sensing inhibitors [61]. The molecular targets common for tannins and flavonoids are presented in Figure 2. 

Oligomeric proanthocyanidins inhibit interleukins IL-1β, IL-6, IL-8, PLA_2_, lipo- and cyclooxygenases 5-LOX, 15-LOX, COX-1, COX-2, inhibit the activation of NF-κB, as well as the secretion of IgG [20]. Plants rich in tannins have various traditional applications, where some of them are already scientifically confirmed, but some remain uncovered by research [62].

Proanthocyanidins (PACs), the condensed tannins, are plant offense and defense molecules which have many human health benefits. They have a broad spectrum of activity, including antioxidant and antimicrobial effects [63]. PACs represent the main compounds of many different edible fruits and berries. Their best sources are cranberries, blueberries, black currants, black chokeberries, or black elderberries [64]. An astringent character is typical for raw persimmon, banana, or carob beans. A high content of PACs was detected in fresh chokeberries, rose hips, and cocoa products [65,66]. 

The extract of blackberries (*Rubus eubatus* cv. “Hull”) rich in PACs can reduce the metabolic activity of various common oral bacteria. It was the most effective against *P. gingivalis, F. nucleatum.* and *S. mutans,* in the concentration range from 350 to 1400 μg/mL [67]. The antimicrobial activity of a lingonberry or a mountain cranberry (*Vaccinium vitis-idaea* L.) was evaluated against two oral pathogens, *S. mutans* and *F. nucleatum*. Lingonberry juice concentrate was prefractionated over reversed-phased resin into fractions enriched with polyphenols. The anthocyanin and procyanidin primary fractions were the most efficient against *F. nucleatum* (MICs from 63 to 125 μg/mL), and the procyanidin-rich fraction against *S. mutans* (16–31 μg/mL) [68]. Duarte et al. (2006) also studied the effect of phenolics isolated from cranberries on the virulence traits of *S. mutans*. Flavonols (125 µg/mL) and PACs (500 µg/mL) alone or in combination, inhibited GTFs (30–60% inhibition) and F-ATPases activities and the acid production by *S. mutans*. Furthermore, the biofilm development and acidogenicity were significantly affected [31]. 

Feng et al. (2013) studied the PACs from cranberries (*Vaccinium oxycoccos* L. cv. ꞌStevensꞌ), for their inhibitory activity against bacterial adhesion and their ability to disrupt cariogenic biofilms. In the study, A-type PACs were used over a saliva-coated hydroxyapatite biofilm model, twice a day for 60 s. The biofilm accumulation was impaired, and the specific genes involved in the adhesion of bacteria, glycolysis, and acid stress tolerance were negatively affected. The results showed that cranberry-specific oligomeric PACs might effectively disrupt the formation of cariogenic biofilms of *S. mutans* [69]. According to Philip and Walsh, cranberry A-type PACs showed potent inhibitory effects against cariogenic virulence targets such as bacterial acidogenicity, aciduricity, glucan synthesis, and hydrophobicity. Cranberry polyphenols can disrupt these cariogenic virulence properties without being bactericidal, which is necessary to maintain the benefits of a symbiotic resident oral microbiota [70]. The topical applications of PACs (1 min exposure, twice daily) significantly reduced the dry weight and the total amount of extracellular insoluble polysaccharides of *S. mutans* biofilms (35–40% reduction compared with the control). It is known that insoluble exopolysaccharides are essential for adhering, coherence, and accumulating microorganisms on the tooth surface. However, the PACs did not affect the accumulation of intracellular polysaccharides in the biofilms [71]. Furthermore, cranberry polyphenols, including PACs, significantly reduced the acidogenicity of the biofilms compared with those of only vehicle-treated [31]. The effect of cranberry polyphenols on streptococci was also investigated by Yamanaka-Okada et al. (2008). It was found that cranberry polyphenolic fraction significantly decreased the hydrophobicity of *S. sobrinus* and *S. mutans* in a dose-dependent manner. The concentrations needed to inhibit the biofilm formation were lower than 500 µg/mL [72]. Kim et al. tested the effect of cranberry polyphenols on the biofilm formation of *S. mutans*. PACs and myricetin were able to inhibit the activity of GTFs and exopolysaccharides (EPS) mediated bacterial adhesion without killing the organisms. The topical application of an optimized combination of PACs oligomers (100–300 μM) with myricetin (2 mM) twice daily was used to simulate the clinical treatment regimen. Treatment with cranberry polyphenols effectively reduced the level of insoluble EPS (>80% reduction) and prevented the outgrowth of *S. mutans* [73].

Dimer PACs such as epicatechin-dimer B-2, catechin-dimer B-3, catechin-epicatechin-dimer B-1, and catechin-epicatechin-dimer B-4 are presented in dry fruits of *Vaccinium myrtillus* L. [20]. The berry fruits (extracts and fractions) exhibited antibacterial effects against various bacteria [74], including periodontopathic bacteria *P. gingivalis*, *F. nucleatum*, *P. intermedia,* and *S. mutans,* with MICs 26 μg/mL, 59 μg/mL, 45 μg/mL, and >62.5 µg/mL, respectively [75]. 

Dutreix et al. studied red raspberry fruit, known for its richness in tannins, as a potential anti-biofilm agent against *Candida spp.* They examined four different extracts from the frozen ripe and unripe raspberry fruit. The most active was the ethyl acetate fraction from the ripe fruit, in which the HPLC/MS analysis identified eight compounds of hydrolyzable and condensed tannins that may be responsible for *C. albicans* eradicating activity. Their work highlights the preventive potential of *Rubus idaeus* L. in oral cavity infections caused by fungi [76]. 

The activity of selected tannins (gallic acid, ellagic acid) was tested against four *S. mutans* strains. With an MIC of 12.5 μg/mL, ellagic acid was effective and inhibited the growth of all tested strains, while gallic acid at the same concentration inhibited only one tested strain [54]. 

One of the most famous middle-European traditional medicinal plants is *Agrimonia eupatoria* L. The aerial part of agrimony is particularly rich in tannins (up to 11%), such as catechin, procyanidin B3, agrimoniin, and other phenolics (astragalin, cynaroside, hyperoside, isoquercitrin, isovitexin, and rutin). Thanks to its astringent, antimicrobial, anti-inflammatory, and wound-healing properties, this herbal remedy is recommended for the symptomatic relief of mild inflammation of the mouth and throat [20,77,78]. According to Ham and Kim (2018), four extracts of an agrimony herb (methanol, water, 50% ethanol, and 95% ethanol) inhibited the *S. mutans* biofilm formation in a dose-dependent manner [79]. Ellagitannins such as agrimoniin (Figure 3), laevitagin Bm, laevitagin F, pedunculagin, and oligomeric PACs are tannins presented in *Potentilla erecta* L. rhizome, which the EMA also recommends for the symptomatic treatment of minor inflammation of the oral mucosa [80]. Methanol extract of Tormentil rhizome in mucoadhesive dosage forms (hydrogel) inhibited cariogenic *S. mutans* biofilm at a final extract concentration of 2 mg/mL in a porcine buccal mucosa model in vitro [81]. 

The North American traditional plant, also well known in European medicine, is the Virginian witch hazel (*Hamamelis virginiana* L.), used to treat minor inflammation and infections involving epithelial tissues (skin and mucosal). Australian researchers indicated that the methanolic and aqueous extracts of *H. virginiana* leaves inhibited the growth of *Staphylococcus epidermidis*, *Staphylococcus aureus, Streptococcus oralis,* and *Streptococcus pyogenes,* with MICs in the concentration range from 200 to 500 μg/mL. *Ex adverso*, *S. mutans* was not susceptible to any of the extracts tested, and the combinations of extracts with conventional antibiotics failed to yield beneficial interactions [82]. The effectiveness of commercial mouthwash containing *H. virginiana* extract on tooth biofilm was tested in vivo. The bacterial plaque index was significantly reduced after 7, 14, and 21 days [83].

Traditional medicinal plants that are rich in tannins could be found in the genus *Quercus* L. Bark from *Q. robur* or *Q. petrea,* are recommended for the symptomatic treatment of minor inflammation of the oral mucosa [84]. Tannins in oak bark include gallotannins, ellagitannins, and catechins. Ellagitannins are here presented by castalagin, roburins A-E, and vescalagin, and PACs of catechin type are presented by dimers (+)-catechin-(4α→8)-(+)-catechin and (-)epicatechin-(4β→8)-3-galloyl-(-)-epigallocatechin [20]. These compounds are responsible for the astringent taste and have anti-inflammatory and antibacterial effects. Oak bark is a part of the traditional and commercial herbal mixtures intended for healing the mouth and teeth [84]. Tannins (gallotannin) were also found to inhibit human salivary α-amylase; therefore, they are suggested to prevent dental caries [85]. Oak bark is not the only product used in phytotherapy. Another traditional oak product used for oral infections is the galls of *Quercus infectoria*. The efficacy of methanol and acetone extracts of galls against *S. mutans*, *S. salivarius,* and two anaerobic gram-negative bacteria *P. gingivalis* and *F. nucleatum,* was examined by Basri and co-workers (2012). The MICs of methanol and acetone extracts were 0.16 and 0.63 mg/mL, and MBCs were 0.31–1.25 mg/mL and 0.31–2.50 mg/mL, respectively [86]. The ethanol extract of *Q. infectoria* galls inhibited the growth of *S. mutans* in a concentration of 125 μg/mL (MIC) and it eradicated *S. mutans* in a concentration of 500 μg/mL (MBC) [87]. The extracts of tannin-rich medicinal plants are important ingredients of plant-based oral care products such as toothpastes and mouthwashes [88]. Oak bark is closely connected to the maturing of red wine. It is generally known that tannins and other phenolics in oak have important functions in aged wines. Reportedly, moderate red wine consumption has proven benefits on human health [89] and is also considered to protect the oral cavity from the cariogenic action of *S. mutans*. Wine contains many phenolic substances, including flavonoids, stilbenes, hydroxybenzoates, anthocyanins, or condensed tannins. Their amount depends on different factors, especially the grape variety and weather conditions [90]. It was shown that dealcoholized red wine strongly interferes with the *S. mutans* adhesion to sHA beads (promotes its detachment from sHA) and therefore, inhibits biofilm formation. Biofilm inhibition was proven on the occlusal surface of natural human teeth as well. The main components responsible for such activities were found to be PACs [91]. 

*Camellia sinensis* L. (green tea) is one of the most popular plants in the medicine and food industry. The leaves contain polyphenols, especially tannins, such as catechin, epicatechin (EC), epicatechin gallate (ECG), and epigallocatechin gallate (EGCG) proanthocyanidins. EGCG (Figure 4) is one of the most abundant polyphenols in the tea and is regarded as the most important pharmacologically active component. Green tea extract has a moderate and broad inhibitory effect on the growth of many types of pathogenic bacteria, including strains of *Staphylococcus* spp., *Streptococcus* spp., *P. gingivalis*, *Prevotella* spp. (Table 4). It was demonstrated that green tea could significantly lower bacterial endotoxin-induced cytokine release. Part of the antibacterial activity may be a selective antiadhesive effect, while green tea inhibits the pathogen adhesion to cells [92]. Rats with experimental periodontal inflammation were treated with the topical application of a green tea catechin-containing dentifrice. Inflammatory cell infiltration in the periodontal lesions were reduced to a greater degree than the control dentifrice at 8 weeks. The gingiva in which green tea catechin-containing dentifrice was applied also showed a lower level of hexanoyl-lysine expression (a marker of lipid peroxidation), nitrotyrosine (a marker of oxidative protein damage), and TNF-α (an indicator of proinflammatory activity) at 8 weeks compared to gingiva in which the control dentifrice was applied [93]. In a randomized controlled trial, 66 healthy human subjects rinsed with a green tea extract for 1 min three times a week, and at the end of 4 and 7 days, there was a significant reduction in *S. mutans* and lactobacilli in the green tea group [94]. Green tea mouthwash was evaluated in controlling the pain and trismus associated with acute pericoronitis compared to CHX mouthwash. Ninety-seven patients with acute pericoronitis underwent debridement and received 5% green tea mouthwash (study group), or 0.12% CHX mouth rinse (control group). Pain (visual analogue scale; VAS), the number of analgesics, the maximum mouth opening, and the number of patients with trismus were determined. The mean VAS score and number of analgesics of the study group were statistically lower than that of the control group between post-treatment days three and five [95]. Green tea consumption for three months as an adjunct to nonsurgical periodontal therapy significantly improved the clinical parameters of mild to moderate chronic periodontitis in a double-blind, randomized, placebo-controlled trial (N = 120). The mean level of antioxidants measured in gingival crevicular fluid and plasma increased significantly in the green tea group, whereas no significant change was seen in the controls. The overall percentage improvement in the gingival index was markedly better with green tea (1.67-fold greater reduction than in controls) [96].

Assam, *Camelia sinensis var. assamica* (J.W.Mast.) Kitam. leaves infusion is a popular drink with an antibacterial effect on different bacteria, including *S. mutans.* [102,103]. The antibacterial activity of green tea against primary bacterial oral cavity colonizers, such as *Streptococcus mitis* or *S. sanguinis,* is attributed to the group of catechins, particularly EGCG, gallocatechin gallate, EC, and ECG [104]. EGCG damaged the cell wall of the gram-positive bacteria by binding to the peptidoglycan through hydroxyl moieties and thus, eliminated the function of a major structural unit necessary for life [105]. Abdulbaqi’s team investigated the synergistic anti-plaque effect of the combination of Assam tea and *Salvadora persica* extract. They evaluated this activity on a mixed suspension of *S. mitis*, *S. sanguinis,* and *A. viscosus*. Apart from the fact that these bacteria form normal oral microbiota, they also play an important role in forming dental plaque. This combination of medicinal plants showed a synergistic effect with a fractional inhibitory concentration of 0.75 [106]. Taketo Kawarai et al. (2016) compared the anti-biofilm efficacy against *S. mutans* of Assam tea and Japanese green tea. The former has more potent activity in the suppression of *S. mutans* biofilm formation. Polysaccharides, which support biofilm formation, are present in higher concentrations in Japanese green tea extracts compared to Assam tea, where a higher amount of galloylated catechins are present, which have an inhibitory effect against the glucan-dependent biofilm formation of *S. mutans* [107].

Tannin-rich plants such as *Rhus coriaria* L. and *Punica granatum* L. were effective against five common oral bacteria in the study provided by Iranian researchers. Both *R. coriaria* and *P. granatum* water extracts had significant antibacterial properties against all tested bacteria (*S. sanguinis*, *S. sobrinus*, *S. salivarius*, *S. mutans*) and were able to inhibit the bacterial biofilm formation on the orthodontic wire. Further investigations are recommended for the widespread clinical use of this extract. *R. coriaria* was most effective against *S. sobrinus* (MIC = 390 µg/mL) and *P. granatum* against *S. sanguinis* (MIC = 625 µg/mL) [108,109]. The antibacterial activity of *P. granatum* peel methanol extracts against oral bacteria *S. mutans, S. sanguinis,* and *S. salivarius* was evaluated by Abdollahzadeh and co-workers, who found that the effective concentrations were 8 mg/mL and 12 mg/mL [109]. Pomegranate peel covers 60% of the fruit content flavonoids, PACs as well as minerals (Ca, Mg, P, K and Na) [110]. The antimicrobial activity of pomegranate peel glycolic extract (PGE) against the periodontal pathogen *P. gingivalis* was studied by Gomes et al. (2016). The researchers used an in vivo model of *Galleria mellonella* larvae for antibacterial testing. The recorded effective concentration of PGE ranged from 2.5 mg/mL to 12.5 mg/mL [111].

An ethnomedicine-relevant plant is *Garcinia mangostana* L., a native Indonesian plant known as mangosteen, which is rich in polyphenols (flavonoids, tannins, and anthocyanins), and was studied in terms of the inhibition of *S. mutans* and *P. gingivalis* biofilms production. The mangosteen peel ethanol extracts inhibited *S. mutans* and *P. gingivalis* biofilms in both a time- and dose-dependent manner. Besides antibacterial action, it is suggested as an antibiofilm agent in alternative therapy for preventing caries and periodontal disease [112].

Natural products that are rich in tannins and flavonoids, and are popular in various ethnomedicines in treating oral infections, have confirmed antimicrobial action (Table 5). 

### 2.3. Bioaccessebility of Tannis and Flavonoids

It is known that the therapeutic activities of tannins and flavonoids are limited by their poor bioavailability. In oral infections, the natural remedies (mouthwashes, gels, tinctures, etc.) are mainly applied topically. The topical application has many advantages in comparison to parenteral or oral (*per os*) applications [116]. The active molecules directly encounter the pathogen on the mucosa or teeth surface and exert their antibacterial effects. A problem may arise if the therapy is needed across different regions that are deeper in the oral mucosa, where the penetration of hydrophilic and hydrophobic molecules differ depending on the mucosal surface type (keratinized or non-keratinized) [19]. Tannins, which are high-molecular-weight substances, create non-absorbable complexes due to their binding properties with molecules in the organism, like proteins. Still, some might be absorbable as smaller units after degradation [62]. Flavonoids (aglycones) are also poorly soluble due to their lipophilic character. Their oral bioavailability is variable and limited [116]. The bioaccessibility of flavonoids and tannins might be increased by modern delivery systems such as nanoencapsulation, using biocompatible and biodegradable materials that are considered safe for humans. Although, the non-toxic properties of such a formulation, and free natural molecules, have to be evaluated by further research [117].

## 3. Materials and Methods

The data for the following review were collected from scientific databases (PubMed, Science Direct, Google Scholar, Scopus) through a search containing keywords “natural products”, “herbal products”, “dental”, “oral infections”, “bacteria”, “candida”, “flavonoids”, “tannins”, “polyphenols”. The main criterion for selecting suitable compounds was antimicrobial activity expression in the MIC, MBC, MBIC50, or IC50 values, which were used to quantify the effect. A further criterion was the parallel information about the activity of the standard conventional antimicrobial drugs, which effect is known. We chiefly included the publications using the microdilution/macrodilution broth method described by CLSI and EUCAST. Works based on biofilm inhibition or eradication were preferred. We considered works from the last ten years with a preference for the latest research. Earlier research or exceptions from the criteria above were cited only if relevant and necessary for precise formulation. The literature search and examination process was performed by three independent researchers (L.S., E.T., and K.R.) and was then checked, edited, and completed by the fourth person (S.B.F.). From around 300, we selected more than 90 works that formed the basis for this review.

## 4. Conclusions

Flavonoids and tannins are the main polyphenols in plants. Both groups have a wide range of pharmacological activities, including antibacterial and anti-inflammatory properties, making them ideal candidates for treating bacterial infections. Many traditional medicines apply extracts of medicinal plants for the topical treatment of mucosal or skin inflammation and wounds. Research over the last decade has shown that polyphenols can reduce the growth of cariogenic bacteria and modulate bacterial biofilms. Preclinical and clinical studies brought significant results that confirm the proper place of traditional medicinal plants in current medicine. The research of natural products remains a crucial area for the discovery of new antimicrobial molecules. Some polyphenols are considered alternatives to conventional antibiotics or may be used with antibiotics to overcome antibacterial resistance. Our review shows that many flavonoids and tannins, as single compounds or in mixtures as natural extracts, are effective agents against bacteria responsible for dental caries, periodontal disease, and other oral infections, considering their availability, efficacy, safety, and finally, the patient compliance. Among the most effective are prenylated flavonoids, catechins, and procyanidins.

## Figures and Tables

**Figure 1 molecules-28-00158-f001:**
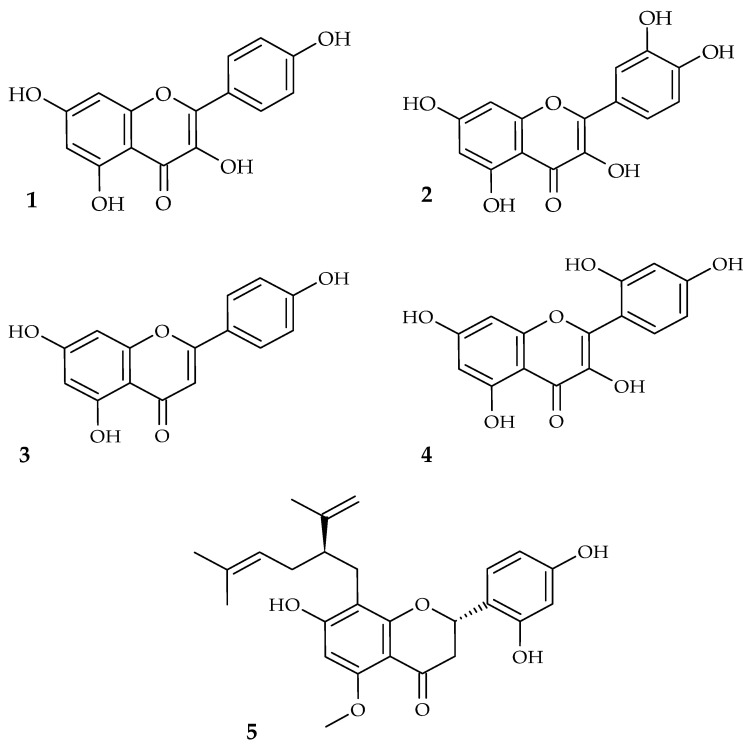
The chemical structures of selected flavonoids with antibacterial activity against agents of oral infections (**1**. kaempferol, **2**. quercetin, **3**. apigenin, **4**. morin, **5**. kurarinone).

**Figure 2 molecules-28-00158-f002:**
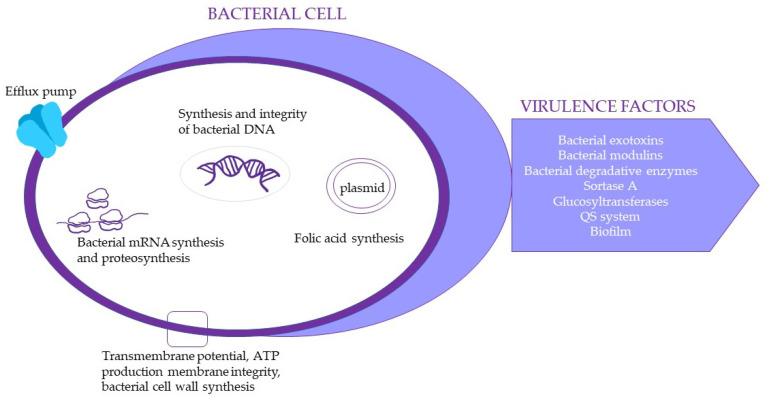
The bacterial molecular targets for tannins and flavonoids antibacterial action [21,29].

**Figure 3 molecules-28-00158-f003:**
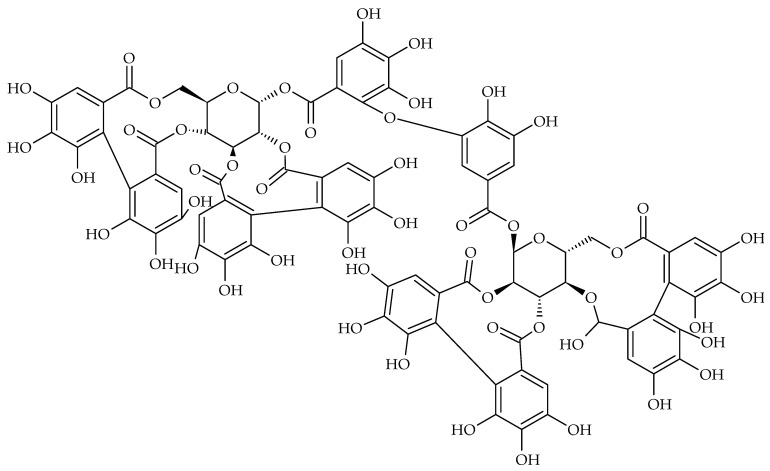
Chemical structure of agrimoniin.

**Figure 4 molecules-28-00158-f004:**
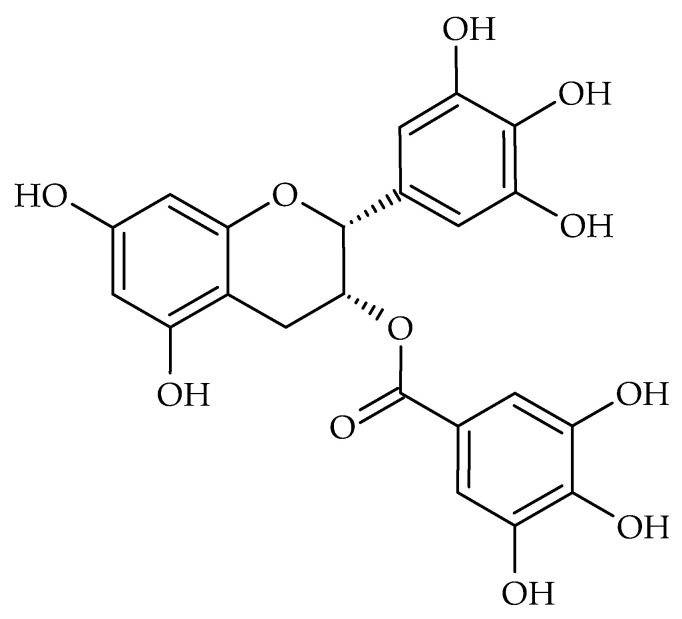
Chemical structure of epigallocatechin-3-gallate.

**Table 1 molecules-28-00158-t001:** The most common oral diseases caused by oral microbiota communities [2,3,4,5,6,7,8,9].

Disease	Underlying Factors	Contributing Oral Microbial Communities	Complications
Dental caries	Xerostomia Sugars-rich diet Insufficient oral hygiene Genetic factors Immunodeficiency	Cariogenic supragingival dental plaque communities Cariogenic subgingival dental plaque communities (cervical and root caries)	Pulpitis, pulp necrosis Periapical abscess, periapical granuloma Dissemination and focal infections, dental sepsis Aesthetic defects and psychological impact
Chronic infection of the dental canal	Improper canal cleaning, shaping and irrigation Insufficient disinfection of the treated dental canal	Non-fastidious members of oral microbiota	Dental canal treatment failure
Periodontal disease	Bad oral hygiene, dental calculus Smoking Hormonal disturbances Genetic predisposition Stress Immunodeficiency	Dysbalanced subgingival dental plaque communities, esp. proteolytic anaerobic bacteria	Tooth loss Chronic low-level inflammation and systemic impact (cardiovascular diseases, Alzheimer’s disease, inflammatory bowel disease, complications during pregnancy) Dissemination and focal infections Halitosis, aesthetic defects, and psychological impact *Cancrum oris*, Vincent`s angina
Oral candidiasis	Impaired local and systemic defence mechanisms Xerostomia Dental prostheses Endocrine disorders (e.g., *diabetes mellitus*) Malnutrition Malignancies Damaged oral mucosa, underlying mucosal diseases Poor oral hygiene Altered or immature oral microbiota (antimicrobial therapy; neonates) Smoking	*Candida* spp. colonizing the oral cavity	Spread into the larynx, pharynx, or oesophagus Disseminated candidiasis

**Table 2 molecules-28-00158-t002:** The most important culturable oral pathogens [8,9,10,11].

Disease	Pathogens	Important Virulence Factors
Dental caries	*Streptococcus mutans,* *Streptococcus sobrinus,* *Bifidobacterium dentium*, *Scardovia wiggsiae,* lactobacilli (*Lactobacillus fermentum*, *L. rhamnosus*, *L. gasseri*, *L. salivarius*, *L. plantarum*, *L. casei-paracasei* group)	Adhesivity, biofilm production (glucans production), acidogenicity—sugar metabolism (acid production), aciduric properties
Chronic infection of the dental canal	*Enterococcus faecalis,* *Enterococcus faecium*, *Candida albicans*, other *Candida* spp., coliforms *Pseudomonas aeruginosa*	Adhesivity, biofilm production, resistance to external factors, proteolytic and cytolytic enzymes, inflammatory potential, antimicrobial resistance, enhanced resistance to disinfectious agents
Periodontal disease	*Aggregatibacter actinomycetemcomitans* *Porphyromonas gingivalis*, *Treponema denticola*, *Tannerella forsythia*, *Fusobacterium nucleatum*	Adhesivity, biofilm production, proteolytic activity and other aggressins, invasion, inflammatory activity
Oral candidiasis	*Candida albicans* *C. glabrata, C. guilliermondii, C. krusei, C. lusitaniae, * *C. parapsilosis, C. pseudotropicalis, C. stellatoidea, * *C. tropicalis*	Adhesivity and biofilm production Proteolytic and lipolytic activity Invasivity Switching to filamentous forms

**Table 3 molecules-28-00158-t003:** Antibacterial/antibiofilm effects of the most abundant flavonoids.

Flavonoids	Bacteria	Antibacterial/Antibiofilm Action	Reference
Quercetin Kaempferol	*S. mutans*	Increasing of the bacterial culture pH. Reduction of the total dry weight of the biofilm. Reduction of the cell viability. Reduction of the formation of insoluble and soluble glucans. Half maximum biofilm inhibition concentration (MBIC_50_ = 16 and 8 mg/mL, respectively), was comparable to chlorhexidine (CHX). Antibacterial activity in concentration 8 μg/mL.	[36,37]
Kaempferol	*P. gingivalis*	Antibacterial activity in concentration 8 μg/mL.	[37]
Rutin Quercetin-3′-*O*-methyl-3-*O*-α-L-rhamnopyranosyl-(1→6)-β-D-glucopyranoside Quercetin	*S. mutans*	Inhibition of sortase A (SrtA) with half maximum inhibition concentration (IC_50_) 134 μM, 186 μM and 2011 μM, respectively.	[38]
Quercetin	*S. sobrinus*, *L. acidophilus*, *S. sanguis*, *A. actinomycetemocomitans* and *P. intermedia*	Antibacterial activity in the concentration range from 1 to 4 mg/mL.	[39]
Apigenin	*S. mutans, S. sanguinis, S. sobrinus, S. ratti, S. criceti, S. anginosus, S. gordonii, A. actinomycetemcomitans, F. nucleatum, P. intermedia*, *P. gingivalis*	Antibacterial activity against cariogenic bacteria: minimum inhibitory concentration (MIC) 25–200 µg/mL, minimum bactericidal concentration (MBC) 100–800 µg/mL. Antibacterial activity against periodontopathogenic bacteria: MICs 100–200 µg/mL, MBCs 200–400 µg/mL. Synergistic effect in combination with antibiotics: 4-fold reduction of MICs of ampicillin or erythromycin and 4–8-fold reduction of MIC of gentamicin.	[40]
Apigenin	*S. mutans*	Reduction in the biofilm total biomass (dry weight), but without changes in bacterial viability. Inhibition of the production of extracellular glucans. Synergy: the combination with *tt*-farnesol and fluoride reduces the acidogenicity of biofilm.	[35,41,42]
Apigenin	*S. sobrinus*	Inhibition of glucosyltransferase (GTF) at the concentration of 1.33 mM, whether the enzyme was in solution (90–95% inhibition) or on saliva-coated hydroxyapatite (sHA) surface (35–58%).	[35]
Apigenin	different streptococci	Inhibition of various GTFs; the IC_50_ in solution were from 58 µM to 98 µM, for the surface absorbed enzymes the IC_50_ was higher (458 µM–1 mM). Modulation of the expression of genes that encode GTFs in *S. mutans* in a planktonic state or in biofilm (c = 0.1 mM to 1 mM).	[34,43]
Kaempferol Apigenin	*S. mutans S. sobrinus*	Inhibition of GTFs at the concentration of 500 µM: in solution (70 to 90%). on the surface (19 to 60%).	[34]
Pinocembrin	*S. mutans*	Growth inhibition; MIC ˃ 500 µM.	[34]
Pinocembrin	*S. sobrinus*	MIC = 250 µM, MBC = 500 µM.	[34]
Myricetin	*S. mutans*	Synergistic effect in combination with *tt*-farnesol and fluoride: reduction of the dry weight of biofilm and total amounts of extracellular insoluble glucans and intracellular polysaccharides. reduction of the expression of glucosyltransferase B in biofilm.	[44]
Quercetin-3-arabinofuranoside Myricetin Procyanidin A2	*S. mutans* *S. anginosus*	Inhibition of the surface-adsorbed glucosyltransferases B and C and F-ATPases at the concentration 500 µmol/L flavonoids.	[45]
Luteolin Morin Naringin Quercetin Rutin	*A. naeslundii, A. viscosus,* *A. actinomycecomitans, E. faecalis,* and *L. casei*	Growth inhibition.	[46]
Morin	*S. mutans*	SrtA inhibition (IC_50_ of 27.2 ± 2.6 μM). Reduction of the biofilm mass (in the concentration of 30 μM).	[47]

**Table 4 molecules-28-00158-t004:** The efficacy of green tea polyphenols against oral bacteria.

Bacteria	MIC	Mechanism	Reference
*P.* *gingivalis*	EGCG (500 μg/mL or 5 mg/mL)	At concentrations above the MIC, established biofilms were disrupted. At concentrations below the MIC, biofilm formation was inhibited.	[97]
*P. gingivalis*	MIC = 250–500 μg/mL	Green tea polyphenols, especially EGCG, completely inhibited the growth and adherence onto the buccal epithelial cells.	[98]
*P. gingivalis Prevotella spp.*	MIC of catechin = 1 mg/mL	Hydroxypropylcellulose strips containing green tea catechin as a slow-release topical delivery system were applied to the pockets of patients once a week for eight weeks. Green tea catechin showed a bactericidal effect in vitro with MIC of 1.0 mg/mL.	[99]
*S. mutans*	EGCG (7.8–31.25 μg/mL)	EGCG showed a dose-dependent inhibition. At sub-MIC concentration (15.6 μg/mL), it significantly suppressed the genes encoding GTFs. EGCG at a concentration of less than 78 μg/mL induced cellular aggregation of *S. mutans*.	[100]
*Eikenella corrodens*	EGCG (MIC_50_ = 0.1–0.25 mM)	Sub-MIC concentration inhibited biofilm formation.	[101]

**Table 5 molecules-28-00158-t005:** Medicinal plants and natural products rich in flavonoids and tannins as therapeutic agents in oral infections.

Medicinal Plant	Extract/Fraction/Material	Microorganism	Activity	Reference(s)
*Agrimonia eupatoria* L.	methanol, water, 50% ethanol and 95% ethanol extracts	*S. mutans*	Antibiofilm	[79]
*Assam tea (Camelia sinenssis var. assamica)*	water extract	*S. mutans*	Antibiofilm	[107]
Chilean propolis	crude extract	*S. mutans* *S. sobrinus,*	Antibacterial Antibiofilm	[56,57,58]
*Garcinia mangostana* L. (mangosteen)	ethanol extracts	*S. mutans* *P. gingivalis*	Antibiofilm	[112]
*Green tea (Camelia sinenssis)*	water, water/ethyl acetate extract	*Staphylococcus* spp., *Streptococcus* spp., *P. gingivalis, Prevotella* spp.	Antimicrobial	[92,113]
*Hamamelis virginiana* L.	methanolic and water extracts	*S. oralis*	Antibacterial	[82]
*Matricaria chamomilla* L.	water extract	polymicrobial	Antibiofilm	[50]
Nidus vespae (honeycomb)	chloroform/methanol extract	*S. mutans, S. sobrinus, S. sanguis, A. viscosus, A. naeslundii and L. rhamnosus*	Antibacterial Antibiofilm	[31]
*Potentilla erecta* L. (rhizome)	methanol extract	*S. mutans*	Antibiofilm	[81]
propolis	isolates	*S. mutans, S. sobrinus*	Antibacterial Antibiofilm synergy	[34]
*Punica granatum* (peel)	crude extract methanol extract in nanoparticles water extract	*Lysinibacillus cresolivorans* *L. boronitolerans* *S. mutans* *S. sanguinis, S. sobrinus, S. salivarius* *P. gingivalis*	Antibacterial Biofilm inhibition	[108,109,111,114]
*Quercus infectoria* (galls)	methanol and acetone extracts	*S. mutans, S. salivarius* *P. gingivalis* *F. nucleatum*	Antibacterial	[86]
Red wine Italian	dealcoholized extract	*S. mutans*	Antibacterial In vitro, ex vivo biofilm inhibition	[91]
*Rhus coriaria* L.	water extract	*S. sanguinis*, *S. sobrinus*, *S. salivarius*, *S. mutans*	Antibacterial	[108]
*Rubus idaeus* (raspberry)	ethyl acetate extract	*C. albicans* *C. glabrata* *C. parapsilosis*	Antiadhesive	[76]
*Salvadora persica L.* (miswak)	Water	*S. mitis* *S. sanguinis* *A. viscosus*	Antimicrobial Synergistic anti-plaque	[106]
*Sophora flavescens* L.	water-ethanol extract	*S. mutans*	Antibacterial	[53]
*Vaccinium oxycoccos* L. or *Vaccinium macrocarpon* L. (cranberry)	flavonoid/proanthocyaidin fractions non-dialysable material derived from cranberry juice	*S. mutans* *S. sorbinus*	Antibacterial Antibiofilm Antiadhesive	[31,69,70,71,72,73,115]
*Vaccinium vitis-idaea L*	juice concentrate	*F. nucleatum* *S. mutans*	Antibacterial	[68]
*Vitis vinifera L.* (seeds)	extract	*P. gingivalis* *F. nucleatum*	Antibacterial Antibiofilm	[55]

## Data Availability

No new data were created or analyzed in this study. Data sharing is not applicable to this article.

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
