# Peer review of "Therapeutic Potential of Flavonoids and Tannins in Management of Oral Infectious Diseases—A Review"

_molecules, 2022, doi:10.3390/molecules28010158_

Round 1
Reviewer 1 Report
In the submitted review paper (molecules-2097805), the authors analyzed the therapeutic potential of flavonoids and tannins (the most abundant polyphenols in plants, as their main bioactive components) for managing oral infectious diseases. Primarily using the results of (numerous) research in the last ten years, they convincingly showed that many flavonoids and tannins (like prenylated flavonoids, catechins, and procyanidins) are effective agents against bacteria responsible for dental caries, periodontal disease, and other oral infections, taking into account their availability, efficacy, safety and the patient compliance.
I enjoyed reading this manuscript; congratulations to the authors on an excellently chosen and implemented topic. The article is very well written and will be helpful to a wide readership.
Although the essence (certainly more critical for the scientific paper) is fully satisfied, the form should be polished to a "high gloss"! Therefore, minor (technical) objections must be sent to the authors for correction.
1. Numerous references need to be adequately cited. Add missing doi markers in literature citation numbers: 9, 14, 17, 39, 40, 55, 65, 74, 90, 106, 109, 110, 111 and 112. Check the complete correctness of citations for 7, 102 and 112!
2. Punctuation marks (excess, more often lack, e.g. dots or spaces) should be checked and/or corrected in sentences (parts of the text) on lines: 30, 82, 215, 237, 309, 427, 472, 476, 504, 507, 525, 561, 607, 618, and 646.
3. It is always better to use italics for Latin words and/or expressions on lines: 146, 434, 437, Table 1, and 540.
Author Response
Dear reviewer, thank you so much for the time you paid to review our manuscript, as well as for all your suggestions on how to improve the text and polish it to a high gloss ?
- Numerous references need to be adequately cited. Add missing doi markers in literature citation numbers: 9, 14, 17, 39, 40, 55, 65, 74, 90, 106, 109, 110, 111 and 112. Check the complete correctness of citations for 7, 102 and 112!
Done, but in some references, the doi information was not available.
- Punctuation marks (excess, more often lack, e.g. dots or spaces) should be checked and/or corrected in sentences (parts of the text) on lines: 30, 82, 215, 237, 309, 427, 472, 476, 504, 507, 525, 561, 607, 618, and 646.
Done.
- It is always better to use italics for Latin words and/or expressions on lines: 146, 434, 437, Table 1, and 540.
Done.
Reviewer 2 Report
Therapeutic potential of flavonoids and tannins in management of oral infectious diseases – a review
Flavonoids are most found in fruits, herbs, stems, cereals, nuts, vegetables, flowers and seeds, while tannins are polyphenol compounds derived from wood, bark, leaves, roots, fruits, and seeds of several plants and can be classified into four different categories viz. condensed tannins, hydrolyzable tannins, phlorotannins, and complex tannins. Oral health is an essential component of the general wellness of the human body, and oral microbiota-associated diseases have a local health impact. The chemical nature of flavonoids and tannins are the backbones of drug design. It is an important research topic. However, I have some suggestions and corrections to the article that are appended below.
Comments:
Point 1: Abstract is a good overview of the topic.
Point 2: Introduction: It is too descriptive and monotonous and has much redundant information. There is a need to re-write the introduction.
Point 3: Data collection: This review is not revealed the search strategies, inclusion and exclusion criteria and risk of bias assessment for individual studies; therefore, there is a need to add a material and methods section.
Point 4: There is a need to discuss the possible molecular targets for flavonoids and tannins
Point 5: May provide a figure to show the schematic representation of the key signaling pathways involve and mechanism of action.
Point 6: Result and Discussion: I think the main weaknesses of the manuscript are the lack of synthetic character and the lack of original interpretation of the available data. It is too descriptive, monotonous and has a lot of redundant information.
Point 7: There is a need to add a subsection to discuss the risk, safety and bioavailability related to flavonoids and tannins.
Author Response
Dear reviewer, thank you for the time you paid to review our manuscript, as well as for all your suggestions on how to improve the text.
Comments:
Point 1: Abstract is a good overview of the topic.
Thank you
Point 2: Introduction: It is too descriptive and monotonous and has much redundant information. There is a need to re-write the introduction.
We rewrote the text of the introduction to make it more compact and lose redundant information. We applied two additional tables to make the text more synthetic and summarizing.
Point 3: Data collection: This review is not revealed the search strategies, inclusion and exclusion criteria and risk of bias assessment for individual studies; therefore, there is a need to add a material and methods section.
The material and methods chapter was extended. More details about the review conceptualization were added.
Point 4: There is a need to discuss the possible molecular targets for flavonoids and tannins
The mechanisms of antibacterial action for flavonoids and tannins are summarized in the new Fig 2. and are described in each chapter. Please see it highlighted in the main text.
Point 5: May provide a figure to show the schematic representation of the key signaling pathways involve and mechanism of action.
We added Figure 2 with bacterial molecular targets for natural compounds.
Point 6: Result and Discussion: I think the main weaknesses of the manuscript are the lack of synthetic character and the lack of original interpretation of the available data. It is too descriptive, monotonous and has a lot of redundant information.
We tried to comb the text to make it more compact and lose redundant information. We added a table (Table 3) to make the text more synthetic and summarizing.
Point 7: There is a need to add a subsection to discuss the risk, safety and bioavailability related to flavonoids and tannins.
There is much information about risk, safety and bioavailability in oral (per os) usage of tannins, flavonoids and medicinal plants, but the topical application is not discussed well. The medicinal plants discussed in the manuscript are known for their traditional usage in oral infections. This long-term usage aspect made them acceptable safe. We consider the discussion about risk, allergy aspects of individuals or individual side effects as taken out of context.
The subsection of tannins and flavonoids bioaccessibility was added.
Thank you.
Round 2
Reviewer 2 Report
Most of the suggestions have been incorporated by the authors in the revised manuscript. Therefore, no issue with considering it for publication